# A New Battery Selection System and Charging Control of a Movable Solar-Powered Charging Station for Endless Flying Killing Drones

Essam Ali [1,2,*](https://orcid.org), Mohamed Fanni [1,3] and Abdelfatah M. Mohamed [1,2]

1   Mechatronics and Robotics Engineering Department, Egypt-Japan University of Science and Technology, Alexandria 21934, Egypt; mohamed.fanni@ejust.edu.eg (M.F.); abdelfatah.mohamed@ejust.edu.eg (A.M.M.)
2   Electrical Engineering Department, Assiut University, Assiut 71515, Egypt
3   Production Engineering and Mechanical Design Department, Mansoura University, Mansoura 35516, Egypt
*   Correspondence: essam.ali@ejust.edu.eg or eng.essam@eng.au.edu.eg

**Abstract:** This paper provides a design, a charging control, and energy management of a movable Photo Voltaic (PV) charging station with an Automatic Battery Replacement (ABR) system to enable drones for ongoing missions. The paper represents the first stage of a three-staged project titled Fall Armyworm (FAW) insect killer. The other two stages involve the flight control of drones and detecting and killing FAW insects. Without chemical methods, the project aims to eliminate harmful FAW insects that are rapidly spreading in Africa and Asia. The power source is a hybrid PV system with energy storage devices (batteries and supercapacitors). The maximum power from PV panels is tracked using three different online methods (PSO, IC, and P&O), and the best method with the highest accuracy is selected. The experimental and simulation results approved that PSO is the recommended method used in this project among the studied methods because of its high target reach (about 97%) and low steady-state oscillation (maximum 2.15%). An intelligent energy management system is investigated and designed to efficiently utilize solar power with a constant-current constant-voltage charger for LiPo batteries. A new Battery Selection System (BSS) is designed and verified to efficiently utilize the harvested energy and increase the mission time. The BSS targets to manage the selection of the appropriate battery to charge and control its charging rate. The system performance is tested using MATLAB software. Then, an experimental setup for the system is built to validate simulation results. The results of simulations and experiments proved the reliability of BSS in different operating cases with an efficiency higher than 97%.

**Keywords:** automatic battery replacement; CCCV charging; battery selection system; drone; PV; insect killer; PSO; IC; P&O

## 1. Introduction

Renewable energies, especially solar energy, have an upward penetration in general agriculture fields, particularly in agriculture robots or vehicles. Currently, modern agriculture depends mainly on robots for harvesting, planting, spraying pesticides, and performing unreachable tasks, which save human effort, costs, and time. Our proposed project is concerned with detecting and killing insects harmful to crops in remote farms which have shortage in power sources. The project needs a renewable power source with a continuous flying killing drone. The project was encountered by many challenges. The first challenge is the absence of traditional power sources in the mission location. The second challenge is the time and energy consumed in returning the drone to the charging platform. The third challenge is the time consumed in charging the drone battery after returning to the charging platform, which affects the mission time negatively.

The works in [1,2] provided different methods for intelligently analyzing and simulating the energy of a Li-ion battery used in remote power sources. An Adaptive Neuro-Fuzzy Inference System (ANFIS) model, a dynamic artificial neural network approach based on

Nonlinear Autoregressive eXogenous (NARX) model, and a soft computing technique were used to simulate the system using MATLAB software. Some works provided solutions for the absence of power sources where renewable energy systems were proposed as renewable power sources, representing a solution for the first challenge. Fuel cell and Photo Voltaic (PV) systems had the highest integration in powering the drones [3]. The method in [4] provided the power required for onboard electronic charging systems on an Unmanned Aerial Vehicle (UAV). An energy management system consisting of MPPT, battery management, and power conversion stages was proposed in this work. A high-efficiency aircraft was designed and fabricated in [5] to validate a proposed fuel cell system.

Some works provided solutions for the second challenge, which is the time consumed by UAVs to return to the fixed charging platform. One of the proposed solutions was wireless charging, which could charge drones' batteries during flying without landing to increase the mission time. A UAV was powered from a wireless laser-powered system, where a laser beam was transmitted to charge a UAV during flight [6]. However, the work field was limited in time and mission reach. A design of a quadcopter platform of unlimited flight time with laser power beaming was presented in [7].

Some researchers concentrated on a wireless charging station from renewable energy sources, not from power lines, to overcome the first and the second challenges [8]. The project in [9] proposed an automatic system for charging the UAV consisting of a module of power supply, charger module, and control module, which could control the take-off and return time of the UAV. A renewable solar power source was used for power transfer with a fully automatic charging system to abandon human supervision to make the system completely autonomous. In [10], a PV wireless charging system was designed to allow continuous flying of full autonomous quadrotors. A resonant inductive coupling technique was used for the drone charging application. The presented project in [11] was concerned with overcoming the limitations and capabilities of wireless transmitting the power for some UAV applications. By applying the proposed concept, the quadrotor could be charged in its place. However, it is valid only for particular tasks.

Other projects, such as the one explained in 2017 by Khonji, have proposed solar-powered systems that used energy storage devices such as a battery to store energy [12]. A mobile robot as a portable battery is used to charge from a stationary solar power and then it moves towards the drone for which its battery has been depleted. An inductive wireless receiver was inserted into the drone to perform the charging process. This method had successfully provided a renewable power source, but it was inefficient and time consuming. It is time-costly and energy inefficient because of the charging time and the consumed time for the mobile robot to move from and to the stationary power station.

Some of the previously proposed ideas have succeeded in putting solutions for the first and second challenges facing the drones for ongoing missions. Some problems still appear in these works, such as the cost of transferring the charging power and the time consumed in the charging process. Some researchers had overcome these problems by replacing the depleted battery with charged a one instead of charging it. Fujii in [13] enabled drones to fly continuously by developing an automatic battery replacement system without being constrained by battery power limitations. However, the system depended on a non-renewable power source to charge batteries before replacing the depleted ones; thus, it was not suitable for remote areas with a shortage of power sources. Another new technique is proposed in [14] to make the drone continuously available in the mission location. A system consisting of a certain number of drones and an adequate number of charging stations was used to guarantee that fully charged drones are always available in the mission location to replace drones for which their batteries were depleted. This work used the drones replacement technique instead of a battery replacement system to make a continuous flying system. Unlike this work [14], a new system with a non-renewable energy-powered changer was designed to enable fast hot swaps for charging eight batteries for many vehicles to provide one vehicle system with an endless working time [15].

The main contribution in our proposed system is to propose a solution for the three challenges mentioned above with a newly proposed energy management technique. The system consists of a continuous movable power supply using a movable power station, a renewable energy source for remote areas using solar panels, and a continuous flying drone using an Automatic Battery Replacement (ABR) system. The project is a movable, endless flying, and solar-powered insect killer. To the best of our knowledge, this is the first project combining the three solutions to overcome the three challenges. A new algorithm is proposed to control the start/stop time of the mission while the effects on the system energy and the mission period were studied in our previous work [16]. A new Battery Selection System (BSS) is proposed to achieve the optimal benefit of available solar power in charging the drone's battery, powering the movable power station, and energizing the ABR system. The new BSS can prioritize the mechanical and electrical loads according to excess or shortage in solar energy during the day. BSS can control the selection of the charged battery and maintain the charging level of the battery based on mechanical loads and available solar energy.

The work aims to reduce the number of drones used for insect-killing missions from four drones to only one by automatically replacing the depleted battery. Using four drones without an ABR system needs more control, weight, and cost to design four charging platforms for the four drones. The movement of the charging station with a drone charging on its roof represents a mechanical challenge as well as high power consumption. Using ABR with one drone for the mission is more efficient for power, control, and cost. This means that the system is not only proposed to serve one drone but also a swarm of drones with 25% of the drones that can be used with an ABR system with an appropriate power rating of the installed movable charging station. In this paper, the conducted system is designed to serve one drone for continuous flying. A new system is designed to control the available power from the PV system to charge the highest priority battery; thus, the drone can find a fully charged battery when it lands on the platform. The proposed system uses a battery replacement system to reduce the drone's charging time. Nevertheless, it is different in using a charging selection system to charge batteries based on the priorities of need.

The paper is organized as follows. Section 2 explains the mechatronic system components as well as the timeline and the priority operation of the mission. The task management system and the automatic battery replacement system are detailed in Section 2. Section 3 depicts the charging process of batteries, the design, and control. The new battery selection system is explained in Section 4. Section 5 introduces the experimental setup of the system. By conducing MATLAB simulation and experimental results, Section 6 validates the design and control of the proposed system. Finally, conducting remarks and some future works are summed up in Section 7.

## 2. The Architecture of Overall System

The mechatronic system components are described in Table 1 while Figure 1a shows the layout of the charging and ABR systems. The electrical arrangement of the charging and ABR systems is demonstrated in Figure 1b.

The mobile Seekur robot is the movable station that carries solar panels responsible for charging the drone's battery bank and Seekur robot's batteries in addition to the mechanical loads (motors). The Seekur mobile robot has two Ni-MH batteries that feed the two engines of the movable station and help the super-capacitors in controlling the voltage of the DC-Bus. The drone's battery bank consists of four LiPO batteries. The system has one flying drone and four batteries for powering the drone. The drone's battery lasts for 13–15 min based on mission conditions. The drone's speed during insect detection is about 20 km/h, which enables the drone to fly about 5 km per battery. Mechanical loads include three motors responsible for the ABR system and two BLDC motors used in the Seekur robot's motion.

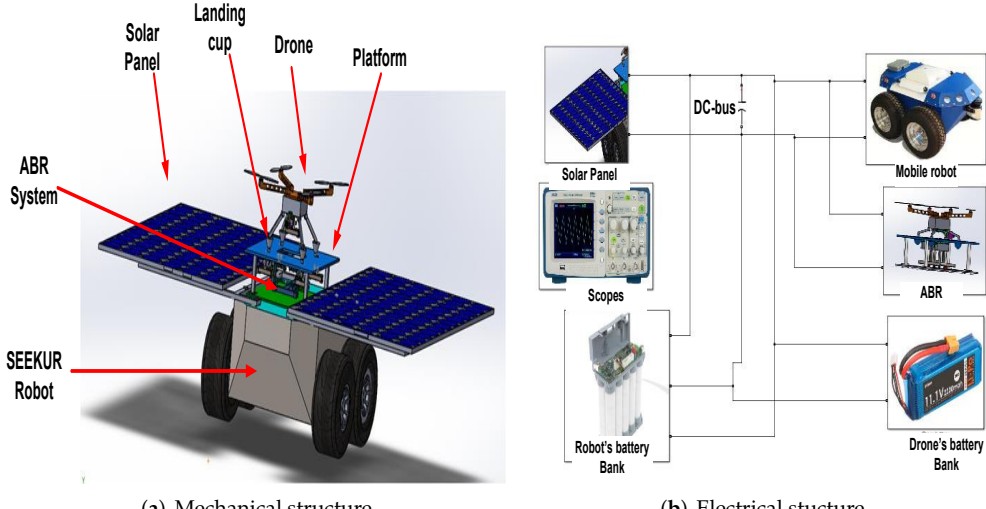

(**a**) Mechanical structure  (**b**) Electrical stucture

**Figure 1.** Mechanical and electrical arrangement of the overall system.

**Table 1.** The ratings of the project's elements.

| Item | Components Details | Ratings | Number | Type | Total |
|---|---|---|---|---|---|
| Seekur Robot | Motors | 320 W, 16 A | 2 | BLDC/BG75 | 460 W |
| | Batteries | 10 Ah, 24 V | 2 | Ni-MH ARTS Energy's Smart VHT | 20 Ah |
| Drone | Motors | 33.6 W, 90 A | 4 | Turnigy T600 | 134.4 W |
| | Batteries | 5.2 Ah, 11.1 V, 35 C | 1 | Jia Long Xing LiPO | 24.42 Wh |
| Solar panel | Electrical Mechanical | 250 W, 17.8 V, 14.4 A $1080 \times 710 \times 2.5$ mm, 2.6 Kg | 2 | OkSolar™ Flexible Monocrystalline | 500 W (max) |

### 2.1. The Timeline of the Mission and Priority of Operation

The drone's mission starts automatically based on the task management system, which will be described in the Section 2.3. The charging time of the drone's battery is limited by using an ABR system detailed in Section 2.4. Some priorities control the mission's timeline to guarantee smooth operation. First, the movable station should not move during battery changes so that the motor's performance is not affected and battery replacement time is shortened. Second, the movable station has the highest priority to be powered, more than the other loads. Third, at least one of the two Ni-MH should be 50% charged before charging the batteries in the drones' battery houses to feed the mobile station motion. Fourth, in the case of the availability of 50%SOC for one of the Ni-MH batteries, the charging priorities will be for the drones' batteries, and the internal priorities for these chargers are governed by BSS, which is discussed in Section 4.

Figure 2 shows the timeline of the mission for one hour in a day. At the first of the day, the EMS controls the mission start, and then the priority operation mentioned above is applied by the energy management system. EMS and BSS are parts of the energy management system. The shown figure shows the energy consumed by the mechanical (ABR + movable station's motors), the electrical loads (Ni-MH, LiPO batteries), and the generated energy from the PV system.

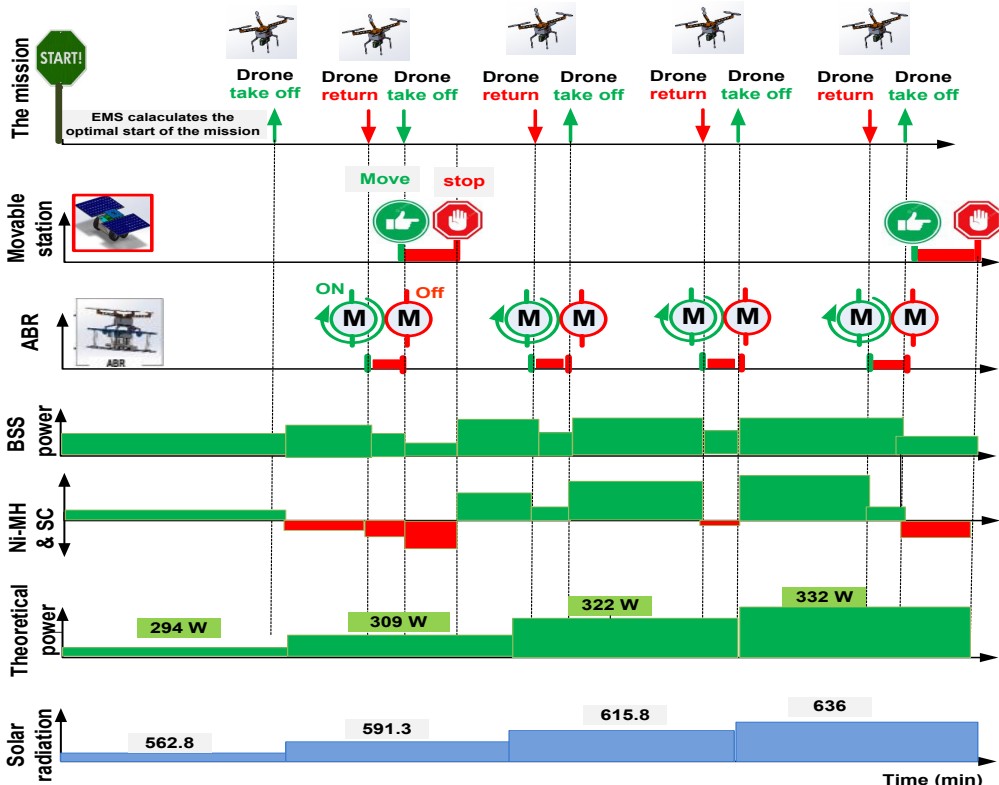

**Figure 2.** The timeline of the overall system.

## 2.2. Deign of the Battery Banks

The PV panels used in the system powering are flexible panels type with maximum power of 250 W. The PV module is 2 × 1 with 500 Watt as the maximum power at standard radiation and temperature. The drone's battery lasts for about 13 min. The drone's automatic battery replacement takes one minute to finish the task [13]. Four batteries are needed for the drone to fly continuously for each hour. In addition to the battery carried by the drone, there are three other batteries suspended by battery bank holders. Each battery can be charged in about one hour using a fast charger (1 C charging rate as recommended).

## 2.3. The Task Management System

The idea of insect detection with a thermal camera is investigated by using a thermal camera in the E-JUST lab where corn leaves covered the insect, and the shown result in Figure 3a was hopeful. As a validation, a 7 W laser beam was used in E-JUST mechatronics LAB in killing the insect where the laser gun was 30 cm away from insect coordinations (Figure 3b).

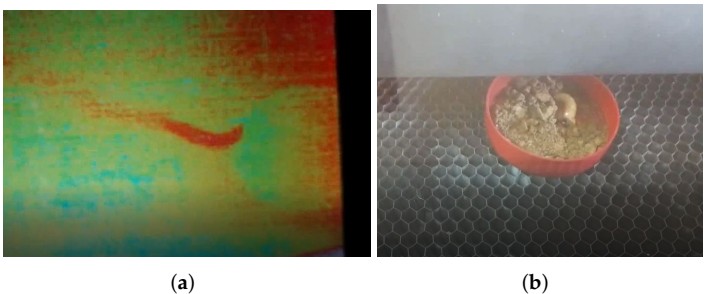

(**a**)                    (**b**)

**Figure 3.** Verifying the idea of detecting and killing the insect. (**a**) Detecting the insect using a thermal camera. (**b**) Killing the insect using 7 W laser beam.

To make the system entirely autonomously controlled, the mission and start/stop times are handled using a task scheduling algorithm shown in Figure 4. If the daily start/stop time for all FAW detecting–killing missions or other missions is fixed on all working days, the power system may not fully charge the next day's batteries for some days since sunset and sunshine times are different. Thus, an electricity shortage can occur at the start of the mission, resulting in an incomplete task. If the charging system failed to charge any of the two Ni-MH batteries from the previous day due to any reason, the Task Management System (TMS) would force the mission to wait until one of the batteries is charged. On the other hand, if the two batteries were charged from the previous day, the task can be hastened early based on the collected PV power predicted by the TMS. Fixing the start time of the mission can cause a loss of power and time during the mission.

Figure 4 shows the flowchart for the algorithm used in time scheduling of the system mission in each day separately, while solar radiation and temperature readings for one day as a sample of each month during this year are shown in Figure 5. The algorithm makes the project a more autonomous and energy-efficient FAW killer. The left part of the flowchart shown in Figure 4 is performed during mission-off times and predicts working hours and start/stop times, while the right part is performed during mission-on times and measures the actual operating hours of the system during the day. The predicted irradiation and temperature data are fed to the system from [17] as a weather predictor. Then, a calculation with the MATLAB system (Figure 6) is used to obtain the curve of the theoretical MPP for each reading (every 15 min reading) fed by the weather predictor [17]. After that, the expected working hours and start–stop times are calculated and saved to the system.

The right side of the flowchart shown in Figure 4 indicates the steps for the actual start/stop time and the actual working hours. First, the mission will not start if none of the two Ni-MH batteries of the movable station are at least 50% SOC level. If the SOCs are lower than 50%, the mission will not start even if the calculation system on the left side of the flow chart provides a specific time to start. The new start time is registered when the mission begins after one of the Ni-MH batteries reaches 50% of SOC. The mission must end with enough Ni-MH SOC levels to return the movable station to the overnight stay location. The effectiveness and accuracy of the system are verified daily using registered data and expected data.

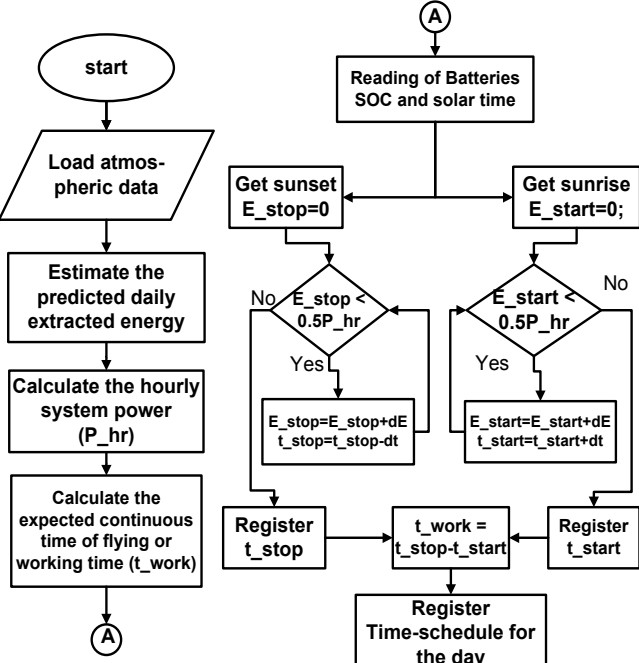

**Figure 4.** The procedure used in calculating mission start–stop times.

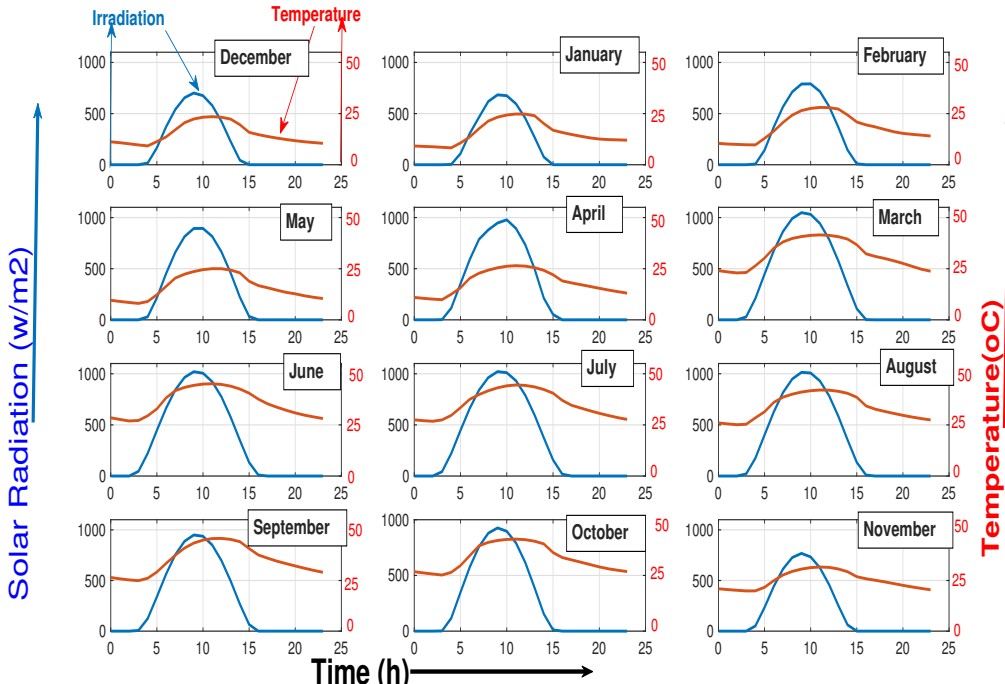

**Figure 5.** Solar radiation and temperature readings in Qena city in Egypt at the first day of each month for one year (December 2020–November 2021).

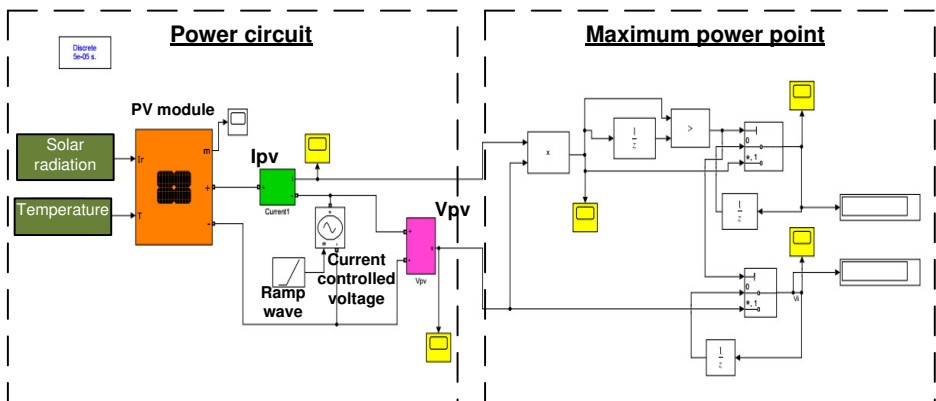

**Figure 6.** MATLAB Simulink for the calculation of the theoretical MPP.

### 2.4. Automatic Battery Replacement

In this work, it is required to minimize the wasted time of the drone mission. Due to the time consumed in replacing the battery and landing of the drone on the charging platform, which may be called "dead time $t_d$," a factor called "Dead Time Factor $K_{dt}$" can be represented by the following equation:

$$K_{dt} = \frac{t_d}{t_d + t_m} \tag{1}$$

$$t_d = t_{ABR} + t_{back} \tag{2}$$

where $t_{ABR}$ is the time of automatic battery replacement of the drone's battery, $t_{back}$ is the time consumed by the drone to return to the station to replace the battery, and $t_m$ is the the mission time taken by the drone to kill the insect.

The $K_{dt}$ factor should be reduced as minimum as possible. The efficient replacement factor ($K_{ER}$) is another factor that can affect the dead time factor ($K_{dt}$), which is the efficient

replacement factor. The $K_{ER}$ factor is an indication for the effectiveness of the ABR system to replace the battery quickly and can be calculated from the following equation:

$$K_{ER} = \frac{t_{docking} + t_R}{t_R} \geq 1 \qquad (3)$$

where the following is the case:

$$t_{ABR} = t_{docking} + t_R \qquad (4)$$

where $t_R$ is the predetermined time consumed to replace the battery, and $t_{docking}$ is the delay time consumed by the docking platform out of the replacement process. $t_{docking}$ can be affected by the movable station's conditions, which either was moving during the drone landing or was stationary. The availability of a fully charged battery may reduce $t_{docking}$ or may increase it until charging one of the suspended batteries in the charging platform. The perfect case is to make this time ($t_{docking}$) equal to zero.

Figures 7 and 8 show the layout of the ABR and the arrangement of mechanical actions from when the drone lands on the robot roof to the moment of taking off, respectively. Firstly, after verifying the correct landing for the proper position and direction, the magnetizing coil is energized to hold the drone's legs while charging the battery. Secondly, ABR system will take the depleted battery off the drone's body and place it inside the empty battery house to be charged. Thirdly, ABR will take the selected fully charged battery to plug it into the drone. Finally, after verifying the complete battery replacement, a take-off command is sent to the drone.

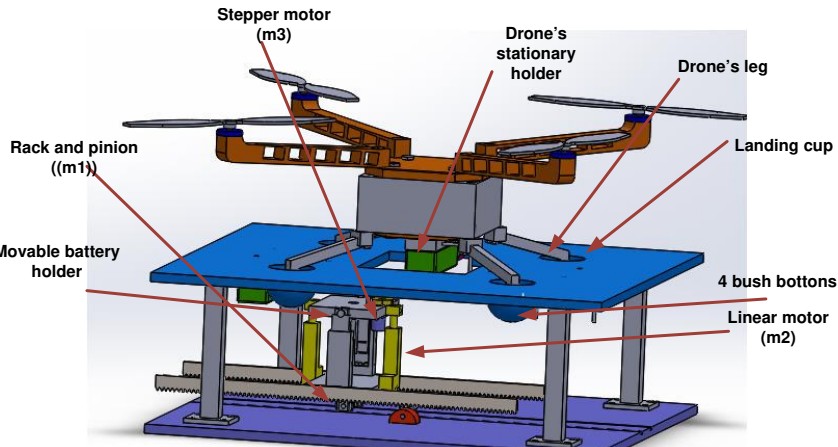

**Figure 7.** The Solidworksvlayout of ABR system.

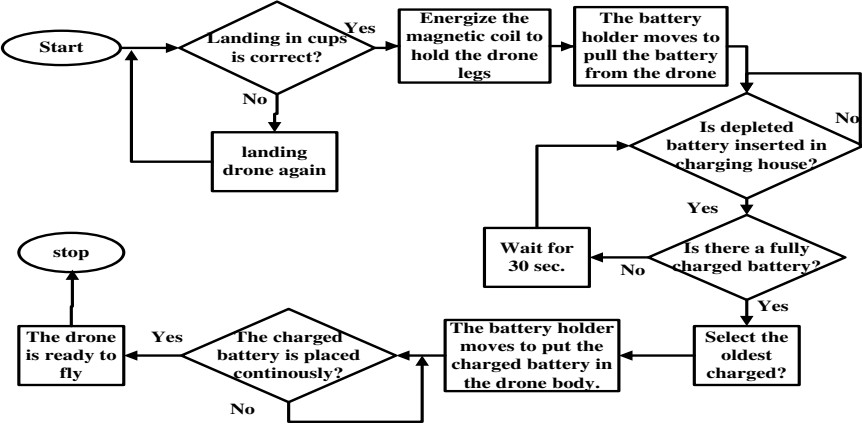

**Figure 8.** The procedure of ABR from the moment of landing to the moment of flying.

## 3. The Charging Process

In this section, the control and design of the charging process of LiPO batteries are proposed. A 5200 mAh LiPO battery is used to power the flying drone because of its high discharge rate, low temperature, and fast charging support [13].

### 3.1. Estimation of State of Charge

Figure 9a shows the charging curve of the 5200 mAh battery with 2A charging current. The figure includes the experimental results and datasheet charging curves. The SOC of the battery is determined using a combined Coulomb method and OCV method to reduce accumulative errors due to the inaccuracy in current measurement by calibrating Ah for each charging cycle. At the beginning of charging, the measured value OCV is used to obtain the initial SOC where the curve fitting method is used to obtain the voltage value, as demonstrated in Figure 9a. Then, the instantaneous SOC during charging is calculated using the Coulomb method (Figure 9b) by measuring current and charging time (measured experimentally by a Real-Time Clock RTC), as shown by the following equation:

$$SOC = SOC_0 + \frac{\int_{t_0}^{t_0+t} I_{bat}dt}{Q_{rated}} \times 100 \tag{5}$$

where $SOC_0$ represents the initial SOC calculated by the extracted curve fitting polynomials, $t_0$ is the beginning of incremental charging, $I_{bat}$ represents the average charging current, and $Q_{rated}$ is the battery's rated capacity (5200 mAh).

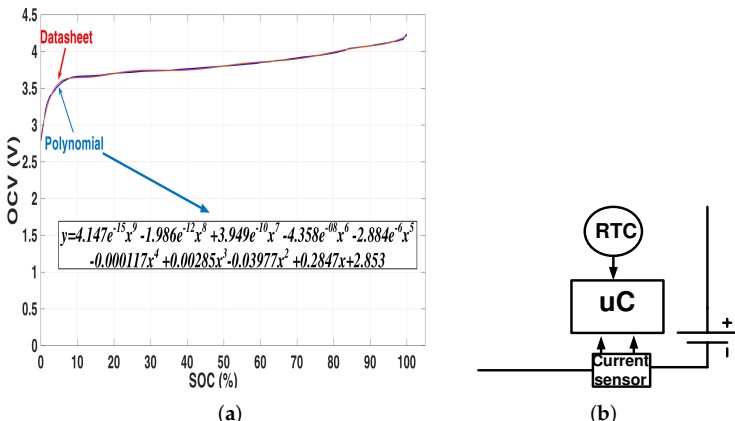

(a)                                        (b)

**Figure 9.** Estimation of SOC. (**a**) OCV vs. SOC from datasheet and experimental curve fitting polynomials. (**b**) Basic diagram of Coulomb counter.

### 3.2. The Safe Operating Area

Li-ion battery causes annoying and dangerous consequences resulting in physical damages if it operates outside the Safe Operating Area (SOA) [18]. Thus, working within the SOA is very important and critical for LiPO batteries (Figure 10). In our design, this issue is well considered by using protection against Over-Voltage (OV), Over-Temperature (OT), Over-Current (OC), Under-Voltage (UV), and Under-Temperature (UT) faults.

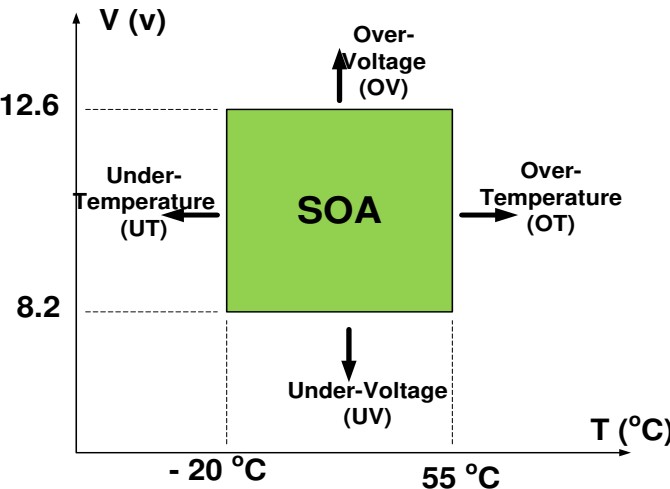

**Figure 10.** SOA of 3S-1P LiPO battery.

### 3.3. Design of Charger

There are different methods used for charging the drone's battery. A 30 V DC-bus is used in our system to feed the mechanical loads and charge the batteries. A buck DC-DC converter is designed to charge the battery using a PID controller. Constant Current Constant Voltage (CCCV) charging technique is designed to charge four liPO batteries. The power and control circuits of the charger are shown in Figure 11. The charger is designed to serve one drone using four batteries replaced by an ABR system.

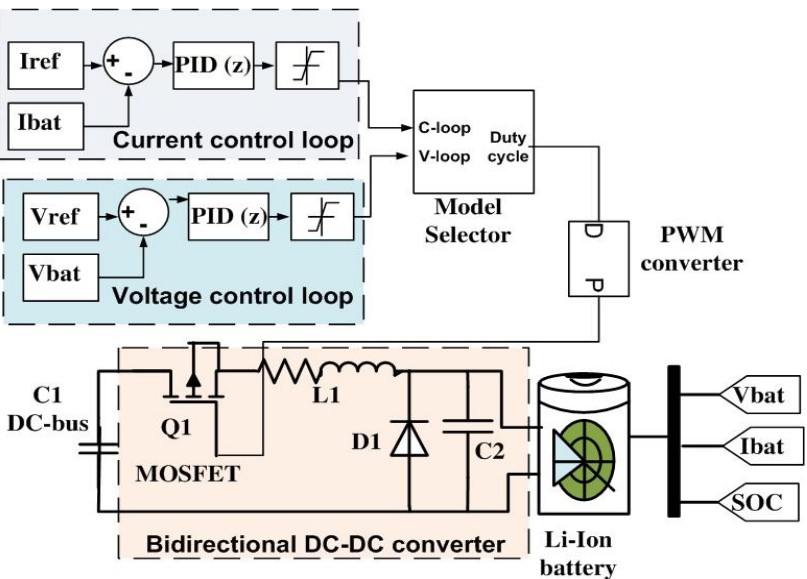

**Figure 11.** The electrical arrangement of CCCV charger in SIMULINK.

The three main designed elements of the converter are the input capacitor ($C_{in}$), the output smoothing filter capacitor ($C_{out}$), and the energy transfer inductor (L). There are two other elements selected after the design of (L, $C_{in}$ and $C_{out}$), which are the switching MOSFET (S) and reverse diode (D). The selection of elements depends on the switching frequency ($f_s$), the output current ($I_o$), and output voltage ($V_o$) ratings of the charger [19]. A 20 kHz-15 V-10 A charger leads us based on the following equations (Equations (6)–(9)) to select a 1.25 mH inductance, 0.1 mF output capacitance, and 0.047 mF input capacitance:

$$L \geq \frac{(1-D).V_o}{2.f_s.I_{rip}} \tag{6}$$

$$C_{out} \geq \frac{(1-D).V_o}{2.f_s{}^2.L.V_{outrip}} \tag{7}$$

$$D = \frac{V_o}{V_{bus}} \tag{8}$$

$$C_{in} \geq \frac{D(1-D).V_{bus}}{2.f_s{}^2.L.V_{inrip}} \tag{9}$$

where $V_o$ and $I_o$ are the output voltage and current. $V_{outrip}$, $V_{inrip}$, and $I_{rip}$ are the ripples in output voltage (selected as 3% of $V_o$), input voltage (selected as 1% of $V_{bus}$), and inductor's current (selected as 8% of $I_o$) and $V_{bus}$ is the voltage of DC-bus (=30 V).

## 4. The Battery Selection System

The battery selection system is a part of the central energy management system, and its primary purpose is to efficiently utilize extracted PV power. BSS can reduce the time of ABR and share and power the drones from the LiPO battery bank equally. This system can manage the battery, which should be charged, and its charging current level. It is expected that the BSS can increase the lifetime of two Ni-MH batteries, which are responsible for charging LiPO batteries in the absence of PV power. The control system must consider some priorities during the selection system, either in the case of abundance or shortage of PV power. The ABR system will not work during the movement of the mobile station. One of the mobile station's batteries (Ni-MH robot batteries RB) must be at least 50% charged before the action of the BSS to feed power to the mobile robot and ABR system if PV power is lost. Distributing the available energy for charging many batteries at a low rate is better for battery life than charging one or two batteries at a high rate. After considering the previous priorities in power feeding, the control of the selected battery to be charged and charging level is summarized in Figure 12.

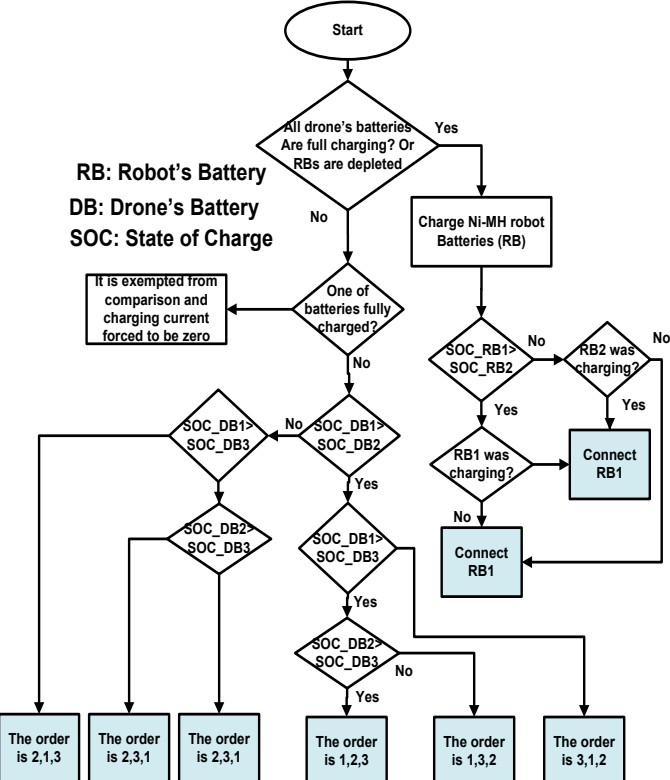

**Figure 12.** The flowchart of BSS.

First, Drone Batteries (DB) are ordered from the highest to the lowest SOC. The time needed for the highest SOC Battery and the remaining time for the drone to return to the platform is compared to balance the availability of a fully charged battery and equal charging of batteries. If the time required for the drone to return is smaller than the time necessary for the battery to charge fully, the battery charging level of the highest SOC battery is increased to reduce the charging time.

Figure 13 shows the PID controller used for selecting the reference charging current for the drone battery bank. The available power is sensed from DC-bus and compared with the charging power for the three batteries; then, a reference charging current is fed to the three chargers.

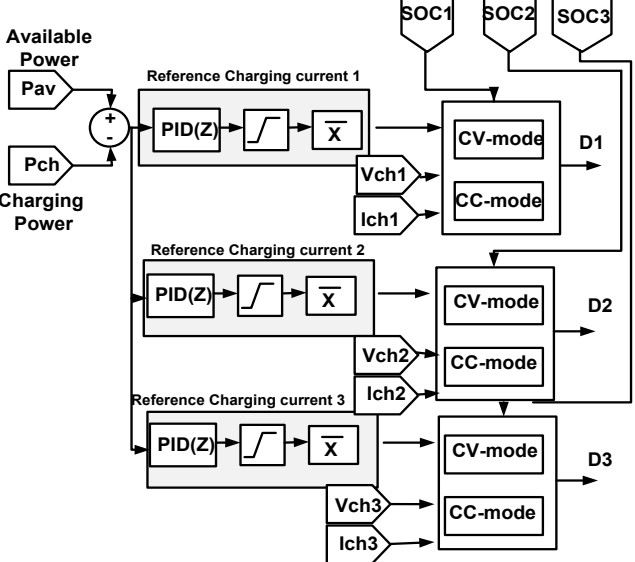

**Figure 13.** The design of BSS controller for the three charged batteries.

## 5. Experimental Setup

Figure 14 shows the experimental setup for the overall system for which its simulation scheme was demonstrated in Figure 1. In this section, an explanation of the experimental structure for the three main parts of this charging system is declared. The three parts are the MPP tracker, CCCV charger, and the BSS.

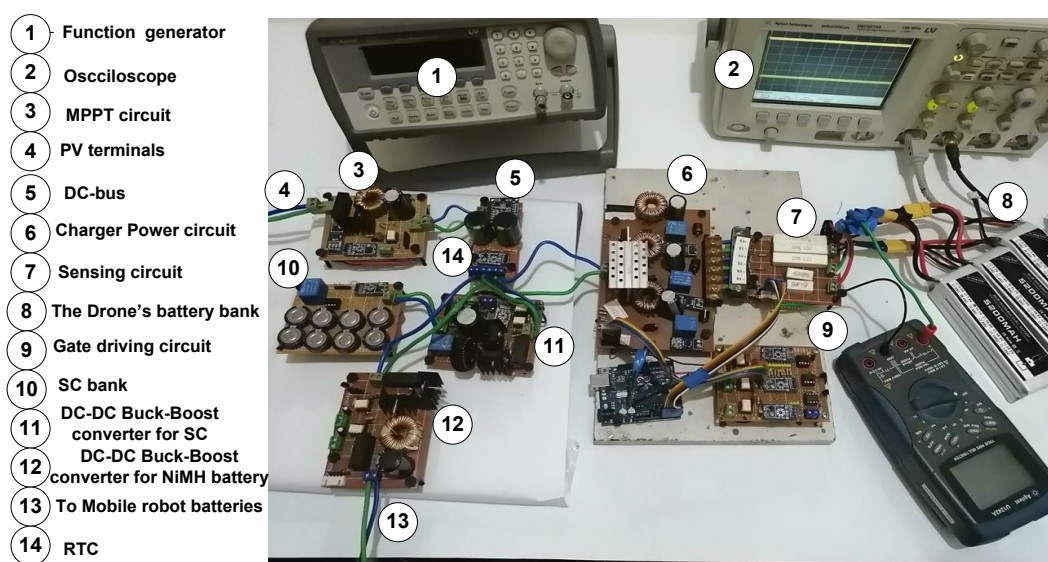

**Figure 14.** The hardware configuration for the proposed system.

### 5.1. Maximum Power Point Tracker

A boost converter with constant output voltage (DC-bus 30 V) is used to track the maximum point of PV power. One panel (75 W, 21.9 OCV) is used to validate the MPPT methods used in this study. A boost converter is used to guarantee that the DC-bus voltage will be higher than PV voltage for all duty cycle states, as shown in Figure 15. Three methods are compared to select the best for running the tracker, Particle Swarm Optimization (PSO) method, Perturb and Observation(P&O) method, and the Incremental Conductance (IC) method. A PWM generator with 20 kHz generates MOSFET gate pulses while the sampling reading time for the experimental and simulation code is 1 ms.

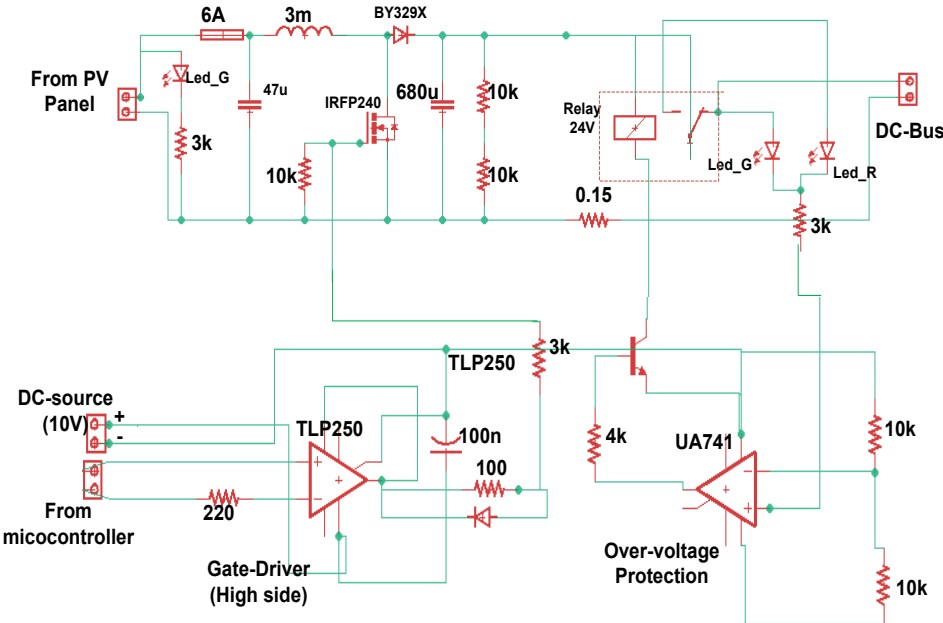

**Figure 15.** The hardware setup of MPPT circuit.

### 5.2. The Charger Circuit

The experimental power circuit for the DC-DC step-down charger is shown in Figure 16, while the gate driving circuit is the same as in the MPPT circuit. The charger is designed with 100 V and 10 A rated power while its input voltage is regulated to 30 V by the DC-bus controller, and a bootstrap fast switching control circuit is used to drive the DC-DC chopper.

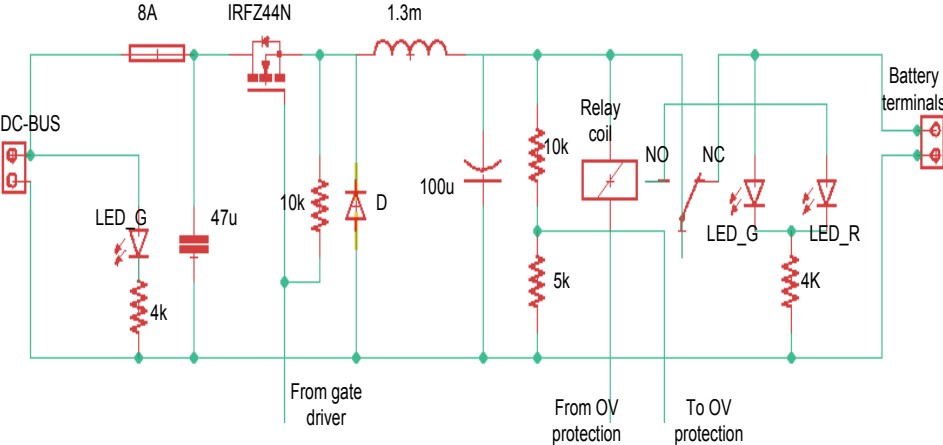

**Figure 16.** The hardware setup of the charger.

### 5.3. Design of the Battery Selection System

The charging platform of the drone's batteries consists of four charging circuits while one of them is not energized, which relates to the empty battery house in the charging platform on the roof of the robot, as shown in Figure 17. The PID controller gains are tuned automatically by using the MATLAB tool. For each battery, there are three controllers, which are the CC controller, CV controller, and charging level controller. In the case of CV mode, the charging level controller is deactivated for this battery and replaced by a CV controller.

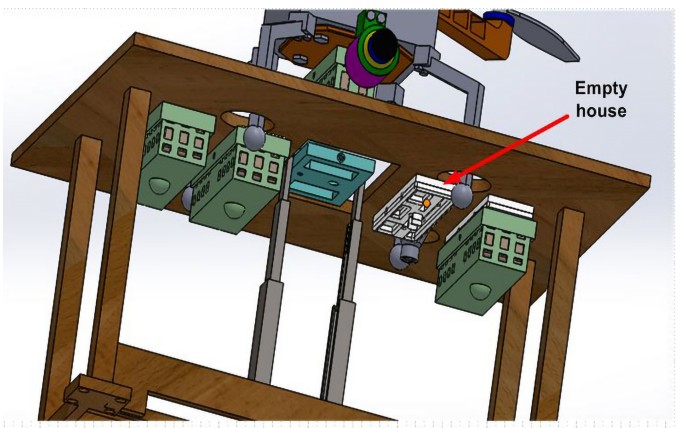

**Figure 17.** The layout of battery houses during ABR.

## 6. Results and Discussion

In this section, a comparison between the simulated system by MATLAB and the experimental setup is made. The results are divided into three main subsections: comparing comparison between the MPPT methods, the charging curves' results, and the battery selection system results. Three methods, IC, P&O, and PSO, are selected to obtain maximum power points due to their easy implementation on the micro-controller system and low execution time. A comparison between the three methods is made to choose the best method for the system to be the MPPT method. The second part of the results section handles the implementation of the CCCV charging technique and studies the comparison between the simulated system and the practical one. The last part of the results demonstrates the effectiveness of the battery selection system in reducing charging time and benefiting the available PV power efficiently. The BSS selects the appropriate battery to be charged and its charging current according to the operating conditions so the system can trade-off between the continuous availability of charged battery for the drone, the equal charge/discharge times for all batteries, and the efficiency of extracting the maximum available power.

### 6.1. MPPT Methods

Three methods are used to extract the PV system's maximum power: IC, P&O, and PSO to choose one of them. Figure 18a shows the simulation comparison between the three suggested methods, while Figure 18b experimentally makes comparisons between them. It is observed that there is a lag in the curve of PSO in simulated and experimental results. The oscillation in P&O is higher than others, while PSO fluctuation is the smallest in simulation and experiment. Although PSO has the slowest rise characteristics (high rise time), the average power is higher.

The extracted energies in a time of 10 s are 453.1J, 441.89 J, and 452.2 J for PSO, IC, and P&O, respectively. From this figure, the theoretical maximum power is 46.6 in the conditions of 680 W · m$^{-2}$ irradiation and a temperature of 48 °C. PSOs lost energy of 14.4 J from the expected 464 J energy in 10 s, while P&O lost 15.1 J and IC lost 25.2 J.

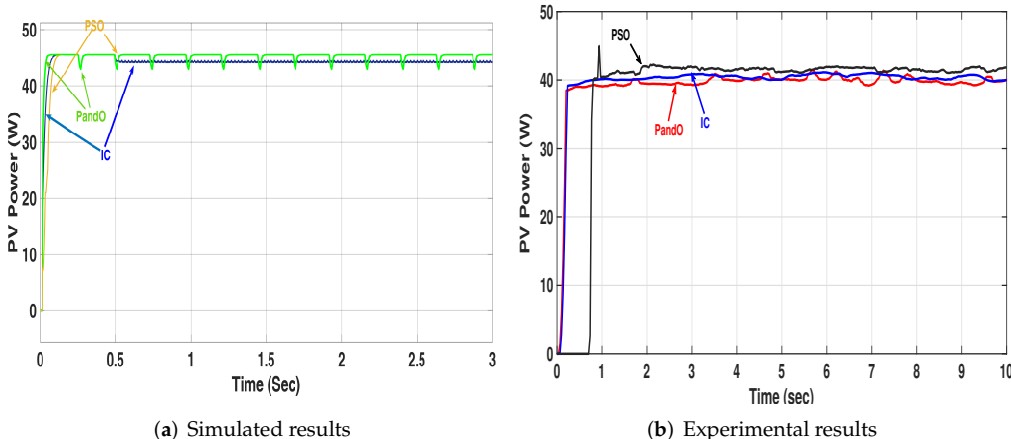

(**a**) Simulated results  (**b**) Experimental results

**Figure 18.** Comparison between different MPPT methods.

## 6.2. Charging Process

The CCCV is the recommended charging technique in LiPO batteries. Figure 19a shows the simulated charging curves of the CCCV charger. Figure 19b shows the curves from the physical charger at the same conditions of 0.5 C charging rate.

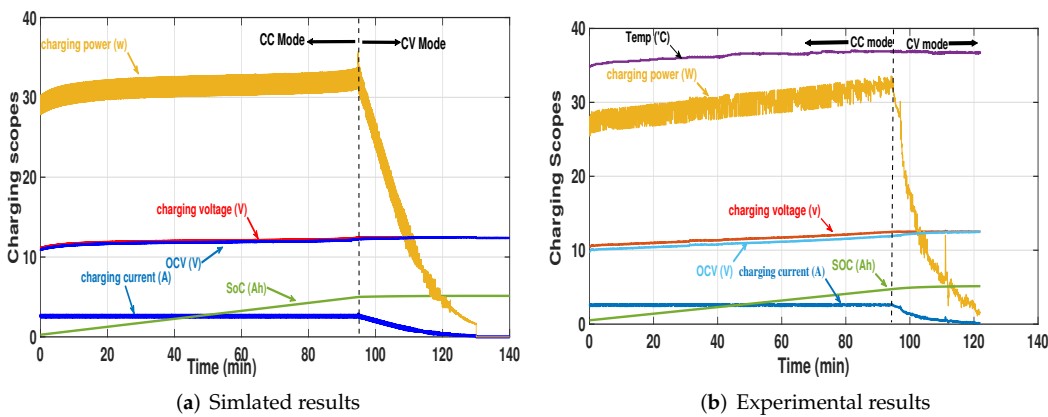

(**a**) Simlated results  (**b**) Experimental results

**Figure 19.** The results curves of the CCCV charging process.

The experimental charging curve begins with 0.5 Ah SOC (about 9.2% of $Q_r$) because it is not recommended to fully deplete the battery while the simulation curve begins with 0.26 Ah (about 5% at the edge of depletion). The experimental curve is slightly different from the simulated results because of the change in controller speed in the two cases. The temperature is sensed by DHT-122 temperature sensor, and as evident, the temperature increases with time. By increasing the charging current, the battery temperature increases. The results shows that the implemented charger is very close to the simulated one.

## 6.3. Control of Battery Selection System

In this section, the results of BSS during different operating conditions in simulation and experimental setup are shown and discussed. The results of BSS are demonstrated in this section to validate the reliability, efficiency, and priority achieved. The reliability of BSS is verified by working on different random operating conditions. The system's efficiency, which indicates the benefit of the maximum available power in charging and powering, is calculated. The priority achievement of the system is verified by changing places of the batteries in the charging houses and observing the system's charging response. BSS followed the the order of batteries discussed in Section 4. During charging, ordering the batteries in battery houses depends on their charge states, while charging current

levels depend on different system variables. The available power, batteries SOC, the time remaining for the drone returning to the platform, and the availability of fully charged battery are variables affecting BSS actions.

Figure 20 shows the simulated current scope of the battery bank for the three batteries during charging, while Figure 21 shows the experimental scope of the current for the same operating condition.

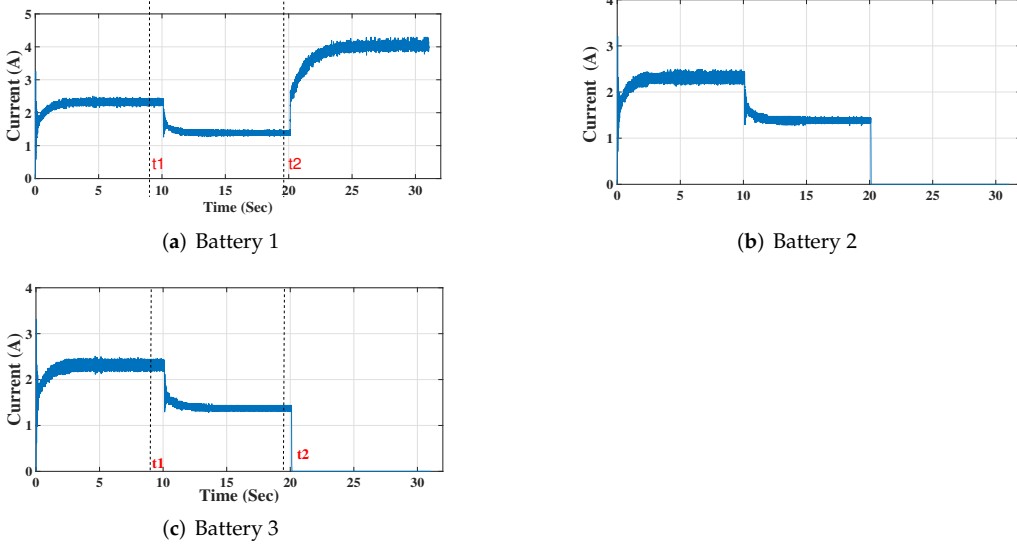

**Figure 20.** Current scopes of batteries for simulated BSS.

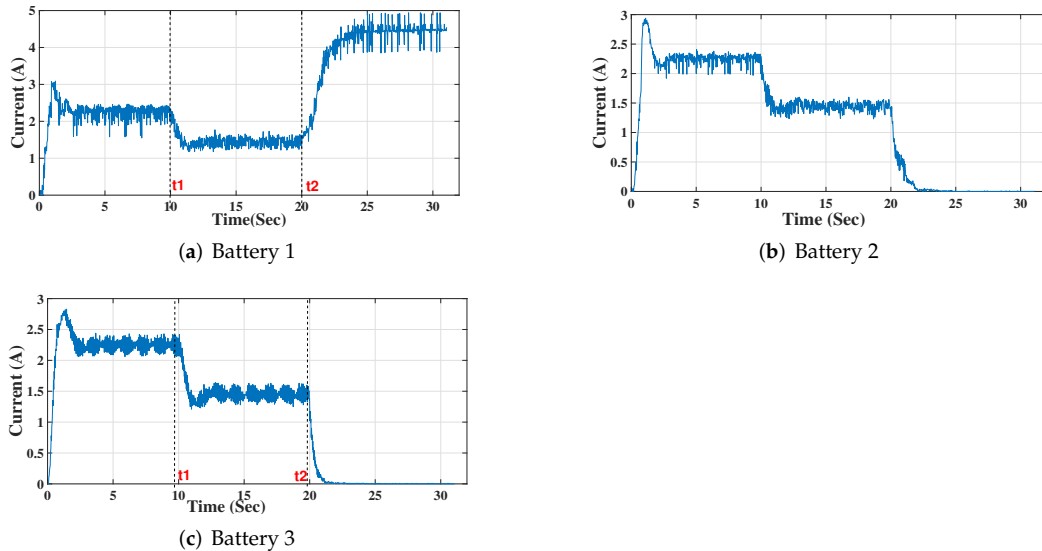

**Figure 21.** Current scopes of batteries for experimental BSS.

In this case, the available power is dropped at time $t_1$ where the three charging currents are reduced to share the newly available power equally. In this case, the calculated time to fully charge one of the batteries is shorter than the time remaining for the drone to return to the platform; thus, power-sharing is equal.

At $t_2$, the time remaining for the drone to return is lower than the time required for complete charging the battery with this charging current. At this moment, the highest SOC battery will charge with its maximum possible current (max 1 C = 5.2 A). The available power is not enough to raise the charging rate to 1 C, but this is considered so that the charging process ends before the drone comes back.

In Figure 22, the highest SOC battery reaches its CV charging mode where its charging voltage is constant at time $t_1$ and its charging current is reduced gradually to zero at the end of the charging process; then, it waits for the drone to return. At time $t_2$, ABR replaced the battery DB1 with the drone's depleted battery DB4. From time $t_2$, the three batteries charge equally with the available power. In parallel to that, BSS calculates the time for the highest SOC—DB2 in this case–to be fully charged to raise its charging level at the right time. At this moment ($t_2$), the SOC of battery 2 (the highest SOC in batteries order) is 92%.

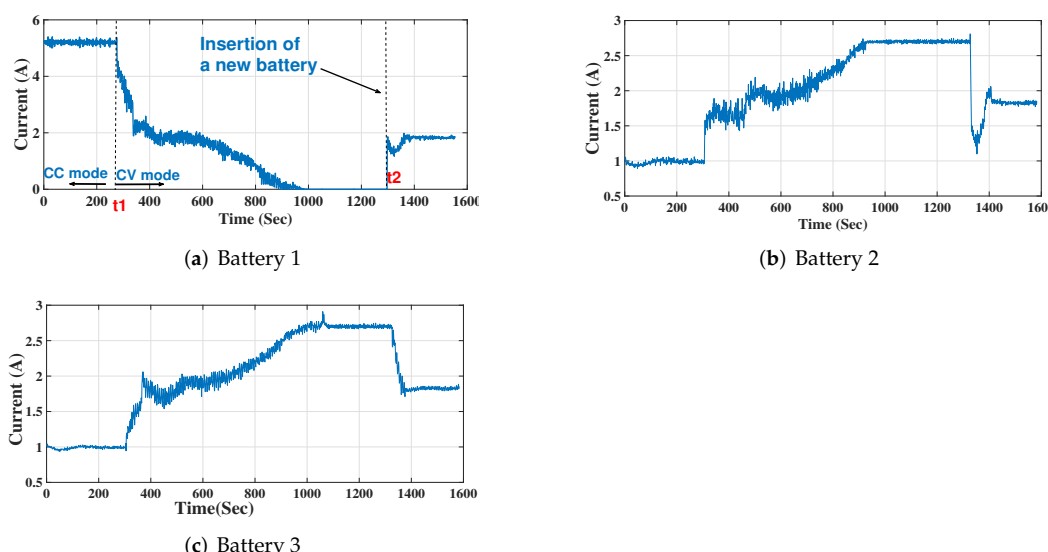

(**a**) Battery 1                    (**b**) Battery 2

(**c**) Battery 3

**Figure 22.** Current scopes of batteries for the experimental BSS in the case of battery replacement.

In the case of low SOC for all batteries, the behavior of BSS charging is shown in Figure 23 where the next charging battery (battery 2) does not have enough time to charge in equal sharing mode. This case is the same as in Figure 22, except that battery 2—which has the highest SOC after replacing battery 1— has low SOC compared to battery 2 in Figure 22. The correct action is to raise the reference charging level of battery 2 (max 5.2 A if possible) at time $t_1$ to reach complete charging quickly. The power mismatch between charging and available power is declared in Figure 24. The curves in Figure 24a–c represent power loss in the previous three operation cases discussed in Figures 21–23, respectively.

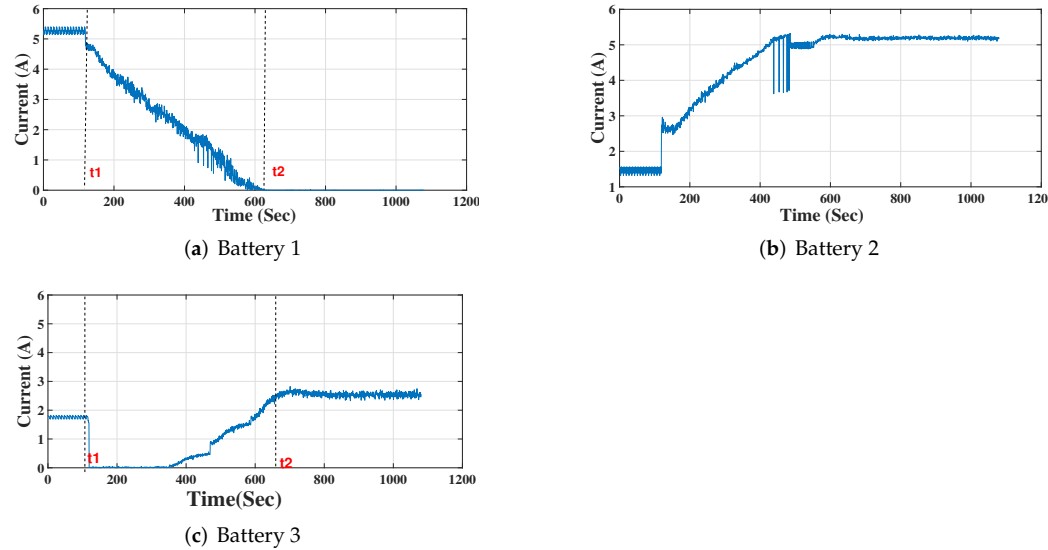

(**a**) Battery 1                    (**b**) Battery 2

(**c**) Battery 3

**Figure 23.** Current scopes of batteries for the experimental BSS in case of low SOCs.

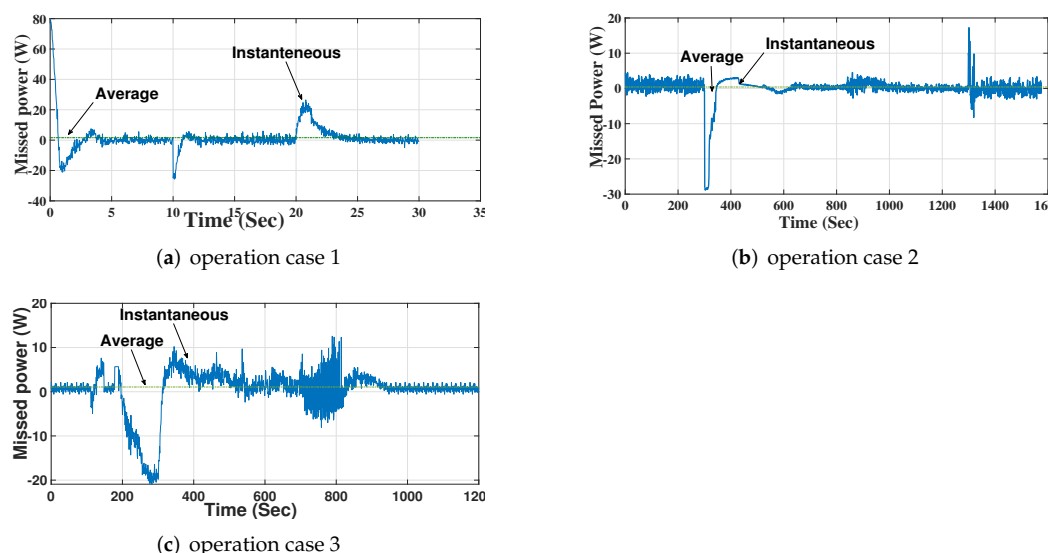

(**a**) operation case 1        (**b**) operation case 2

(**c**) operation case 3

**Figure 24.** Experimental power loss during using BSS.

## 7. Conclusions

A PID CCCV charger was designed and tested with a flexible reference charging current with different protections (OV, UV, OC, and OT). A charging platform consisting of four smart LiPo battery chargers was designed and experimentally tested. A solar MPPT tracker was designed to experimentally verify different MPPT methods in MATLAB simulations. The experimental and simulation results approved that PSO was the recommended method used in this project among the studied methods because of its high target reach (about 97%) and low steady-state oscillation (maximum 2.15%).

In order to efficiently use PV power, we proposed a new battery selection system to control the charging process of batteries and select the charged battery based on the priority code. The BSS controls the flexible reference charging current and changes gradually up to 5.2 A (1 C) based on available power. Both simulated and experimental results highlight the effectiveness of the proposed BSS to select the appropriate battery and charging levels. Different operation cases were studied to verify the reliability of the proposed technique, and the average power loss in different cases did not exceed at the worst case, 1.5 W (or less than 2%).

In the future, a design of automatic battery replacement system can be experimentally built. The target is to limit battery replacement time to 60 s in order to reduce the dead time and increase the mission time of the drone. New methods of MPPT can be used, such as artificial neural networks, ant colony, Cuckoo search, and chicken swarm optimization. A simple Image-Based Visual Servoing (IBVS) algorithm will be used for drone landing.

**Author Contributions:** Conceptualization, E.A., M.F. and A.M.M.; Data curation, E.A. and A.M.M.; Formal analysis, E.A., M.F. and A.M.M.; Funding acquisition, E.A. and M.F.; Investigation, E.A.; Methodology, E.A., M.F. and A.M.M.; Project administration, E.A. and A.M.M.; Resources, E.A. and M.F.; Software, E.A., M.F. and A.M.M.; Supervision, M.F. and A.M.M.; Validation, E.A., M.F. and A.M.M.; Visualization, E.A.; Writing—original draft, E.A.; Writing—review & editing, E.A., M.F. and A.M.M. All authors have read and agreed to the published version of the manuscript.

**Funding:** This research received no external funding.

**Institutional Review Board Statement:** Not applicable.

**Informed Consent Statement:** Not applicable.

**Data Availability Statement:** Not applicable.

**Conflicts of Interest:** The authors declare no conflict of interest.

## Abbreviations

The following abbreviations are used in this manuscript:

| | |
|---|---|
| PV | Photo Voltaic; |
| ABR | Automatic Battery Replacement; |
| FAW | Fall Armyworm; |
| BSS | Battery Selection System; |
| UAV | Unmanned Aerial Vehicle; |
| BLDC | Brush Less Direct Current; |
| TMS | Task Management System; |
| Ni-MH | Nickel—Metal hydride; |
| DB | Drone's Battery; |
| RB | Robot's Battery; |
| SOC | State of Charge; |
| OCV | Open Circuit Voltage; |
| RTC | Real-Time Clock; |
| SOA | Safe Operating Area; |
| OV | Over-Voltage; |
| OT | Over-Temperature; |
| OC | Over-Current; |
| UV | Under-Voltage; |
| UT | Under-Temperature; |
| CV | Constant Voltage; |
| CC | Constant Current; |
| CCCV | Constant Current Constant Voltage; |
| MPPT | Maximum Power Point Tracking; |
| PSO | Particle Swarm Optimization; |
| IC | Incremental Conductance; |
| P&O | Perturb and Observe. |

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
