# Peer review of "A New Battery Selection System and Charging Control of a Movable Solar-Powered Charging Station for Endless Flying Killing Drones"

_sustainability, doi:10.3390/su14042071_

Round 1

Reviewer 1 Report

In this study the concept of drones charging system powered by renewable enrgy sources is proposed. Authors built the special power station for drones and developed the systems of charging the drones batteries to reduce the drones exclusion from work.

The strenght of the paper is the developed stand for charging the batteries powered by PV modules. The Authors contribution is also the proposed algorithm of charging the batteris on the basis of Automatic Battery Replacement. However papers suffers for some lacks.

The major concern is the lack of the optimization assumption and description of assumptions for the charging station composition. Auhtors do not provide how they select the power of the PV modules for charger and how the charger was selected. What was the assumption for charging? How many drones and batteries should be charged by the charger? How long drones oparate on the battery and how long routes (the longer distance) they operate? 

Besides the experimental part is unclear. What was the aim of the experiments? What would you like to achive and show by the results? It is not clearly stated.

Other comments:

  • Please check the grammar and spelling with the native speaker there is a lot of mistakes, for instance: maybe not be (line 132), makses (line 141)
  • The text from lines 97-102 should be moved to the section 2 (also with figure 1),
  • please put the aim of the work in the introduction, 
  • the style correction is needed thorughtout the paper, the lanaguage style is not proper for the scientific text (is rather as report from the task than scientifi article). Please try to improve the style and rewrite the sentences where the words "our project" is used.
  • In section 2, a few word about the tasks that drones do is needed. Please describe what drones do. Where they operate? What is their route and distance they travel? In which part of world they will be used? What is the sunlight conditions in this place?
  • Line 110-122: please add the manufacturers of the batteries and PV modules used in the charging station.
  • The procedure presented in the figure 3 need more explenation in the text. The major steps of procedure should be explained in the text, as well the formulas used for calculations should be provided.
  • Please explain why you tested the three methods to extract the MPP of the PV system: IC, P&O, and PSO? Why it is important for this study, when there are established MPP tracking techniques?
  • What is the achivements presented by the results in the results section? Did you reduce the number of drones without chrged battery? The results of your experiment does not confirm the conclusions and the efficiency of your proposed method. There are just some measurement, however the aim of the experiment is unclear.

Reviewer 2 Report

This paper renders a design, charging control, and energy management of a movable Photovoltaic (PV) charging station with an Automatic Battery Replacement (ABR) system to enable drones for persistent missions. This paper represents the first stage of a three-staged project entitled Fall Armyworm (FAW) insect killer. The other two stages are flying control of drones and detecting-killing FAW insects. It is also mentioned that "the target is to get rid of harmful FAW insects; which are rapidly spreading in Africa and Asia; without chemical methods' '. Regarding the significance of the role of batteries in sustainable planning and smart cities, the idea of the paper is timely. Nevertheless, as a review paper, there are some major issues that need to be addressed as follows:

1- Authors need to prepare a comprehensive flowchart to describe the methods used in the manuscript. 

2- It is claimed that "the BSS system’s goal is to manage the selection of the appropriate battery to charge and control the charging rate. The system performance is checked using MATLAB software. Regarding the limitations of MATLAB, how can the proposed method be used for practical applications?

3- In order to help readers understand the methods verified in this research, it is necessary to add some tables, charts, and graphical works, showing the understudied papers. The quality of figures 9, 13, 16-21 is not clear at all. All figures should be placed in the paper with acceptable quality. 

4- The literature review is very poor. It is highly recommended to use some important publications in the aspect of applications of AI in modeling, optimizing, and improving batteries as follows: An intelligent approach for nonlinear system identification of a Li-ion battery  DOI: 10.1109/I2CACIS.2017.8239040 /b. A dynamic artificial neural network approach to estimate thermal behaviors of li-ion batteries DOI: 10.1109/I2CACIS.2017.8239043/ c. Using a soft computing method for impedance modeling of Li-ion battery current DOI: 10.1504/IJAIP.2020.106686/ d. An ANFIS Approach to Modeling a Small Satellite Power Source of NASA DOI: 10.1109/ICNSC.2019.8743333

5-The reliability, accuracy, and features of the proposed method need to be more discussed. Moreover, the effectiveness of this system should be explained.

6- Authors should show the differences and similarities of the current method in the battery selection system and charging control with other ones with a focus on strengths and weaknesses.

7. Future work should be extended.

Round 2

Reviewer 1 Report

Dear Authors, thank you very muc for provided changes. The manuscript was significantly improved. 

Please revise the text from lines 183-203. You mentioned 3 figures in that part and sometimes you used right and lef side of the graph, but you do not give the references to the graph, it is unclear which figure do you mean.

Author Response

The authors would like to thank the reviewer for the suggestions and comments which lead to improve the manuscript. The text referred in the comments is editted. 

Reviewer 2 Report

All comments have been addressed and the manuscript can be considered for publication. 

Author Response

The authors would like to thank the reviwer for the suggestions and comments which lead to improve the manuscript.